# The Investigation of Interlaminar Failures Caused by Production Parameters in Case of Additive Manufactured Polymers

**DOI:** 10.3390/polym13040556

**Published:** 2021-02-13

**Authors:** Peter Ficzere, Norbert Laszlo Lukacs, Lajos Borbas

**Affiliations:** 1Department of Vehicle Elements and Vehicle-Structure Analysis, Budapest University of Technology and Economics, H-1111 Budapest, Hungary; lukacsnorbert98@gmail.com (N.L.L.); borbas.lajos@edutus.hu (L.B.); 2Engineering Institute, Edutus University, H-2800 Tatabánya, Hungary

**Keywords:** PLA material properties, interlaminal failures, 3D printing parameters

## Abstract

The use of three-dimensional (3D) printing technologies is an ever-growing solution. The product realized in many cases is applicable not only for visual aid, or model, but for tool, or operating element, or as an implant for medical use. For correct calculation, a proper model that is based on the theory of elasticity is necessary. The basis of this kind of model is the knowledge of the exact material properties. The PLA filament has been used to perform this study for matrix material. Our presumption is that the different layers do not fuse completely, and they do not fill up the space available. The failures between the layers and the deposited filaments and the layer arrangement could be the reason for the direction-dependent material properties of the 3D printed objects. Based on our investigation, we can conclude that the increase of the layer thickness and printing speed adversely affect the mechanical properties of the product.

## 1. Introduction

The direct application of products that are manufactured by additive technologies is common [1]. The use of three-dimensional (3D) printing technologies is an ever-growing solution [2]. The product realized in many cases is applicable not only for visual aid, or model, but for tool, or operating element, or as an implant for medical usage [3,4]. For the correct calculation of the load capacity of a certain model, a proper model that is based on the theory of elasticity is necessary. The basis of this model is the knowledge of the exact material properties [5,6]. When the task is the modelling of the original geometry, the form of the object must be kept, and the elastic behavior of the object must be ensured [7,8]. In some cases, the expectation is to realize a deformation that is given to the effect of external load [9,10,11]. In the case of pharmaceutical application, the model can be optimized for different purposes, i.e., the enlargement of the carrier surface, ensuring the necessary biocompatibility [12,13].

The Fused Deposition Modeling (FDM), the filament traction 3D printing technology, is one of the best-used additive technologies [14]. During the procedure, through an extrusion head, a filament heated over its fusing point is put upon the platform, and then on the printed layers (Figure 1):

The binding force between the embedded layers is less strong than the binding force inside the deposited filament [16,17]. The printed material’s loading capacity depends on the quality of the material, the temperature of the printing head, the printing speed, the layer thickness, and the printing direction. Based on the unique build-up of the test specimens, the measurement results were influenced by the type of the test specimen, as it was determined by Amabel García-Domínguez et al. [18,19]. The fatigue properties of 3D specimens were investigated in a study [20] by Sophia Zieman, and it was determined that the direction of the infill pattern has a significant influence on the material properties of the specimens [21,22]. The determination of the inner filling structure—extent and direction—has a significant effect on the mechanical properties [23,24]. Furthermore, we can find cases when the specimens must be designed not only for static, but for dynamic, periodical loading conditions. To this end, the material’s response to dynamic loading conditions must be known [25,26]. The validation of the proper material model using numerical simulation is also necessary [27,28]. Orthotropic theory can describe the material model of the FDM production technology [29]. Complex knowledge of the technology is necessary to optimize the design procedure [30] and ensure the trouble-free utilization of products manufactured in FDM. Unfortunately—for the time being—we do not possess this complex knowledge about this production technology [31]. The material properties produced by this technology must be known for the product’s whole lifetime [32,33]. Our study tries to reveal how the deposited filament’s behavior by the production technology parameters is influenced.

## 2. Materials and Methods

To perform this study for matrix material, a commercial PLA (Polylactic acid) filament—produced by Bq, a Spanish filament supplier—has been used. For this investigation, the exact type of material was not important, since, in our article, just the nature of this phenomenon has been investigated. The properties of different materials can surely influence this phenomenon, but its nature will be the same. PLA is a thermoplastic, holocrystalline polymer that is produced by polycondensation or fermentation of hydroxy propionic acid. It behaves as a rigid material at room temperature, with a tensile strength of 50 MPa, and breaking strain of 3–5%—according to the manufacturer [34]. The reason for the material choice is given in the properties of the material (easy to print and purchase, cheap, no deflection). The specimen’s production was realized by a Bq Hephestos type, open-source FDM 3D printer (easy, at home producible). The free choice of the production parameters was ensured in the case of the printer. 

In this paper, three printing parameters were investigated. Layer height and printing speed have a great effect on surface quality and costs. The smaller the layer height, the longer the manufacturing. Furthermore, they may have an effect on mechanical properties. Temperature is a serious point during polymer manufacturing. Some important properties of polymers, like Melt Flow Index (MFI) and decomposition, depend on temperature, which means that optimum must also be known for high quality 3D printing.

Our presumption is that the different layers do not fuse completely, and they do not fill up the space available, as illustrated in Figure 2. This Figure indicates the failures theoretically occur between the layers and deposited filaments. The failures can be named as porosity, but, since this phenomenon should not appear in a perfect case, it may be better to name it as failures, due to the nature of this phenomenon. This phenomenon and the layer arrangement could explain why the direction-dependent material properties of the 3D printed object are essential. The 3D printers—in general—are equipped with a 0.4 mm diameter extrusion head, through which the fused material flows upon the previous layer [35]. The mathematical models that were established by Bellini et al. explain the pressure decrease in the extruder head [25,36], on the basis of which the volume of the extruded material was investigated. On this basis, in the case of increased printed speed, the extraction force is growing, and the extruded material’s temperature is decreasing [37]. The interlaminar failures are essentially influenced by this phenomenon. The failure dimensions, in many cases, reach some percentage of the filament diameter [38]. To follow and describe this phenomenon, a ratio was determined as a size of the failure that is related to the extruder head (nozzle) diameter. It is the relative failure size (L), as indicated in Figure 2. In this case, *l_failure_* means the maximum size of each gap.
(1)$$L=\frac{l_{failure}}{D_{nozzle}}$$

Figure 2 introduced the concept, being supported by the picture shown in Figure 3. The picture shows a test specimen produced with 0.2 mm thickness. On the marked area, the height/width ratio approximately equals to two, which is a good interpretation of a thickness layer of 0.2 mm that is produced by a 0.4 mm nozzle diameter.

The printing head puts the layer in fused state upon the former layer. Partial fusing state occurs because of this fact. This effect only lasts for a short time, because of the filament dimension the layer cools down very quickly. The period how long the area is heated depends on the velocity of the printing head. In case of the connection of two layers, the molecule segments and chain ends are also connected to each other [39]. The printing temperature, the speed of the printing head, and the layer thickness are the influencing factors of the layer quality. The cooling effect during the printing procedure also has an influencing effect, but, in this study, this effect was not investigated [40]. 

The Ultimaker Cura software was applied to slice the test specimen, in order to generate a code interpretable for the printer. The printing parameters were adjusted by using Cura software, which is indicated in Table 1. Only one parameter was modified during our investigation in one step.

In the case of the specimen design, the size, simplicity, and easy breaking possibility were the points. The connections of the layers and deposited filaments were investigated by Scanning Electron Microscope (SEM). The final form of the specimen was a simple cuboid shape. After breaking, the inner structure was investigated (Figure 4):

After breaking the specimens were investigated with SEM, at the Department of Polymertechnics of BME. Our theory was that the interlaminar failure could be investigated by SEM Technology. The specimens were investigated by CT techniques to avoid the influence of the breaking procedure on the failure. The CT investigation before and after the breaking procedures were carried out at the University of Széchenyi Istvan, Győr.

The distribution of the failures is consistent inside the specimen, independent of the state (broken or not) of the specimen, as shown in Figure 5. The conclusion from this fact is that the interlaminar failures introduced in SEM pictures characterize the whole specimen. Based on CT, the porosity of these 3D printed specimens is high. This phenomenon can be decreased with higher material flow, or with higher temperatures. 

## 3. Results

The effect of the three influencing factors can be determined based on our investigation.

### 3.1. Layer Thickness

Failure increase is observable in pictures produced by SEM technique in the same dimension scale of layer thickness of 0.2 mm and 0.3 mm (Figure 6):

Figure 7 shows the effect of the layer thickness on the relative failure lengths:

As it can be seen, the relative failure lengths with the increase of the layer thickness are increasing and the exponents indicate that the average increase of failure length is 0.75% if the layer thickness is increased by 1%.

### 3.2. Printing Speed

The optimum of the mechanical strength and the production speed is in the same interval. Figure 8 shows the effect of the printing speed on the failure length:

The printing speed has a measurable effect on the size of the failures, as shown in Figure 8. It is important to remark that the speed and acceleration also have a negative effect on the size of the failures.

### 3.3. Printing Temperature

Figure 9 depicts the effect of the printing temperature on the relative failure length:

## 4. Analysis and Discussion

The dimensions of the failure increased linearly with the increase of the failure thickness, as introduced in Figure 7. In this way, we can conclude that the material properties are weakening. In a study, Dénes Tóth concluded that the tensile strength of a specimen manufactured with 0.05 mm thickness layer was about 20% and 30% larger than the ones manufactured with 0.1 and 0.15 mm layer thickness, respectively [41]. For this reason, the nature of this phenomenon can be easily observed. Although the results coming from decreasing the layer thickness meet our hypothesis, the effect of any other influencing factor also has to be taken into consideration. 

Based on this study [41] and our investigation, the tensile strength of the specimen that is manufactured by the indicated parameter can be calculated. These results, of course, must be validated in the near future, although the future results can be well predicted with this investigation.
(2)$$R_{m}=\mathrm{const}=\frac{F_{\mathrm{ijk}}}{A_{\mathrm{ijk}}}$$
where,
i = [0.05 mm; 0.1 mm; 0.2 mm; 0.3 mm; 0.4 mm]j = [200 °C; 210 °C; 215 °C; 220 °C; 225 °C]k = [5 mm/s; 10 mm/s; 20 mm/s; 40 mm/s; 60 mm/s]

The left side of Figure 10 shows that, based on the measurement taken as reference values, the tensile strength of the standard (etalon, with 0.2 mm layer thickness) specimen was the starting point, then the theoretical cross-section values were modified with the failures that were measured at individual layer thickness. The decreased and increased layer thicknesses were 0.1 mm and 0.3 mm, respectively. The results are on the right side of Figure 10. According to the measurement, the slope regression line shows that 0.05 mm of thickness increase would decrease the ultimate tensile force by 30 N. The exact results must be investigated with standard specimens in the future. 

Layer thickness had the most significant effect on the failures. The layer thickness range was between 0.05 and 0.4 mm. The extreme values are not in the practically used range, although the instrument is able to produce them. The 0.1 mm thickness is not visible and, in the case of 0.4 mm, the porosity is unacceptable. Figure 11 shows a SEM picture, where the layer thickness is 0.3 mm; it seems to be the highest thickness from practical point of view [42]. 

The aforementioned study indicates [41] that, if the printing speed is investigated, we can see that the specimen that is manufactured by 50 mm/s production speed is stronger by about 30% than the others, which were manufactured by 150 mm/s production speed. Our results were similar. Anyway, it must be mentioned that the production speed of 150 mm/s is extremely high in the case of FDM technology. Agassant et al. determined that, for FDM technology, the ideal interval for production speed is 40–140 mm/s. Increasing the production speed the width of the extruded material is decreasing; in this way, the size of the interlaminal failures is increasing [38].

Based on our measurement [41], the tensile strength, which was assumed from the failure sizes caused by the different printing speeds, was calculated. In the reference study, different ranges were also investigated, and curve fitting was used in our calculation (Figure 12):

It is important to know that, the higher the speed, the higher the acceleration, which reacts to the accuracy of the model. The uncertainty of mass flow calculation is growing as a result of increasing the speed and acceleration. The regression curve shows that the increase of the printing speed would decrease the ultimate tensile force, according to the measurement. Figure 13 shows the effect of these parameters. 

The temperature effect investigation shows different trends than that introduced previously [44]. If the temperature is higher, an optimum value can be reached. As Figure 14 indicates, the optimum of the temperature is about 215 °C, depending on the manufacturer and the material. The calculated tensile strength as a function of the printing temperature can be seen in Figure 14:

The study of Mouhamadou et al. introduced that, in the case of higher printing speed, the polymer temperature at the outlet of the printing head decreases [38]. The printing head temperature could be corrected based on this fact.

## 5. Conclusions

In this study, three printing parameters of 3D production technology were investigated. The main goal was to investigate the effect of the printing parameters on the structure of the printed element, such as layer and deposited filaments connection. Based on our investigation, we can conclude that the increase of the layer thickness and printing speed adversely affect the mechanical properties of the product. 

These parameters also influence economic production. The necessary production time is increasing linearly while decreasing the layer thickness and the printing speed [45]. Concerning the temperature effect, it can be stated that the printing temperature has an optimum value. Increasing the temperature, the layer connections are fixable; over this temperature, the surface quality could become worse, as it can be seen in Figure 15. 

An additional fact is that, over that optimum printing temperature, some irreversible changes could occur to the material.

Consequently, the smaller layer height can lower the porosity of 3D printed parts, but, at the same time, the production time is increasing. 

In this study, the porosity of 3D printed specimens has been observed. This phenomenon must be exactly determined, and further investigations are required. The variety of materials and 3D printers make it impossible to define the exact or best 3D printing parameters.

## Figures and Tables

**Figure 1 polymers-13-00556-f001:**
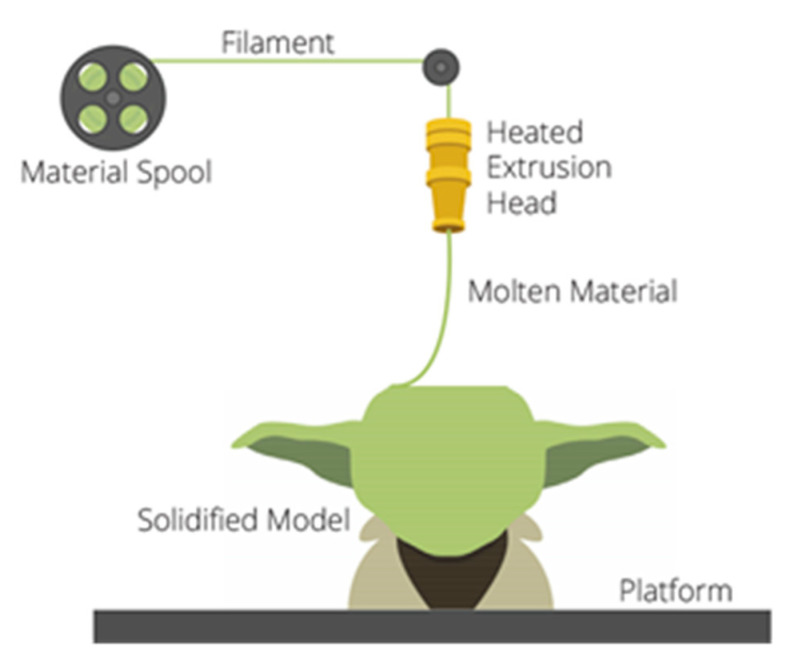
The principle of Fused Deposition Modeling (FDM) technologies [15].

**Figure 2 polymers-13-00556-f002:**
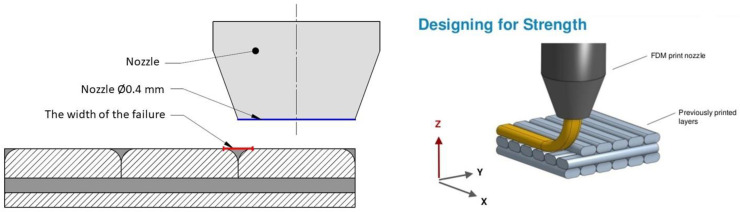
Definition of relative failure.

**Figure 3 polymers-13-00556-f003:**
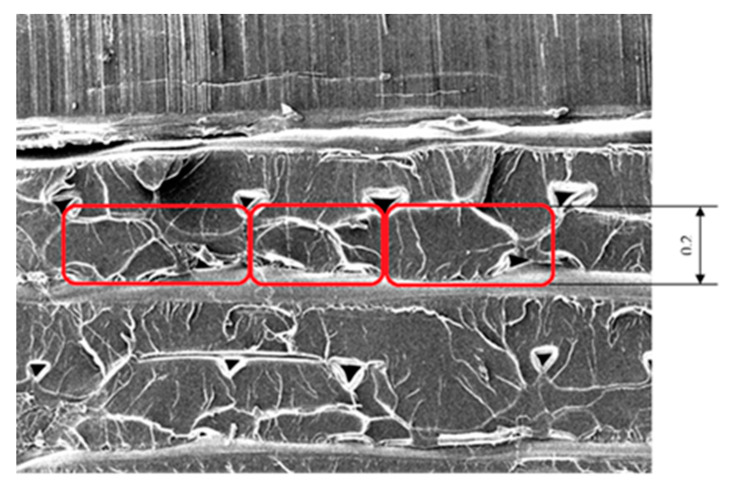
Layer thickness of a specimen of 0.2 mm, produced by a 0.4 mm nozzle diameter.

**Figure 4 polymers-13-00556-f004:**
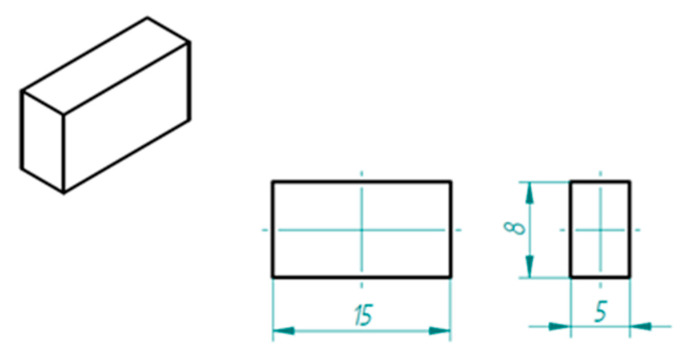
The specimen (please note dimensions are in mm).

**Figure 5 polymers-13-00556-f005:**
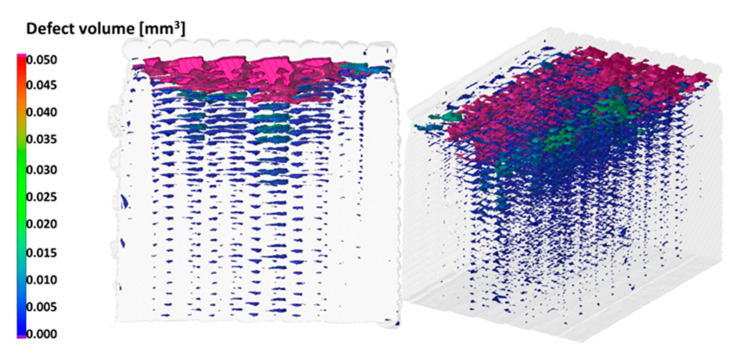
CT picture of a specimen.

**Figure 6 polymers-13-00556-f006:**
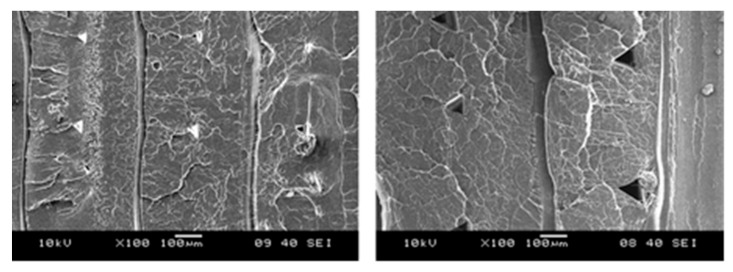
Scanning Electron Microscope (SEM) pictures of layer thickness of 0.2 and 0.3 mm, in the same dimension scale.

**Figure 7 polymers-13-00556-f007:**
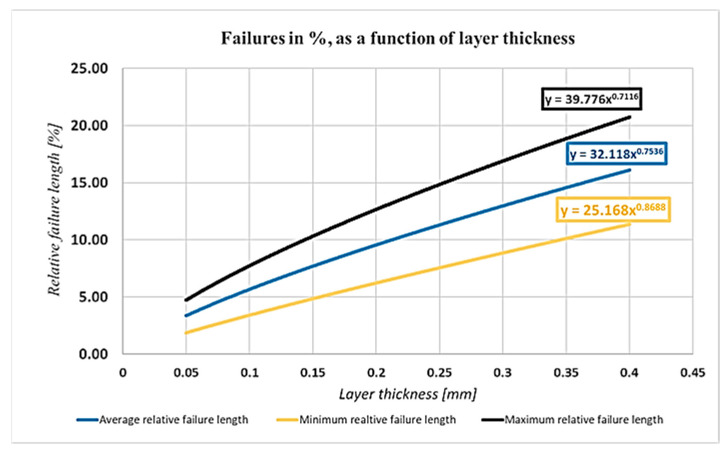
Effect of the failure thickness.

**Figure 8 polymers-13-00556-f008:**
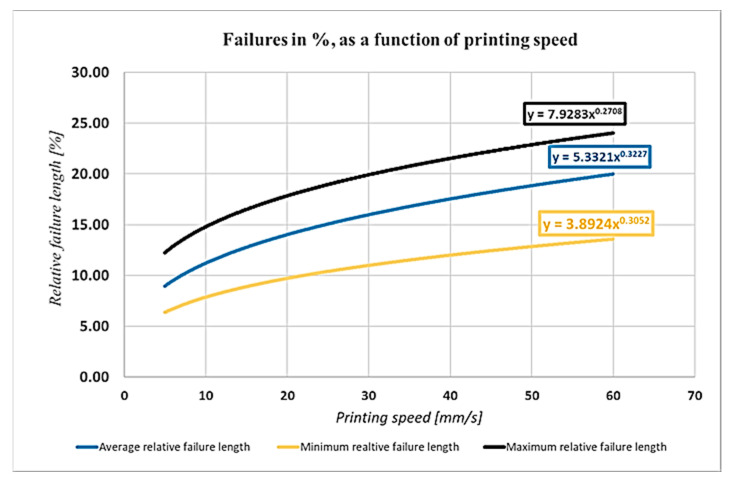
Effect of the printing speed.

**Figure 9 polymers-13-00556-f009:**
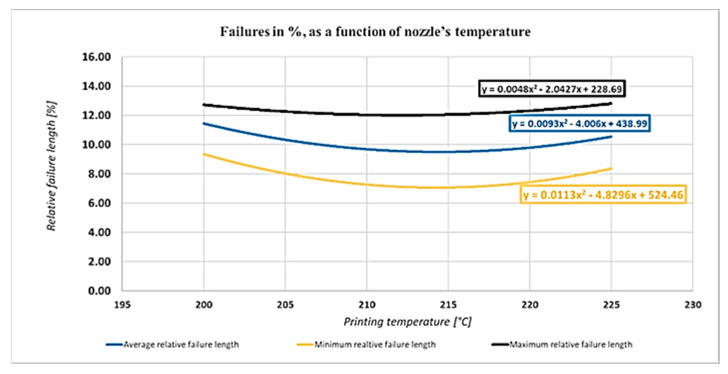
Effect of the printing temperature on the relative failure length.

**Figure 10 polymers-13-00556-f010:**
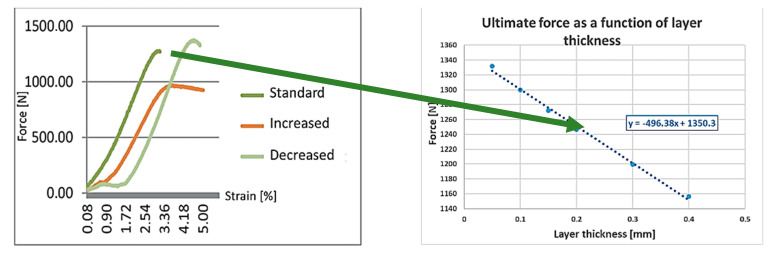
The calculated tensile strength as a function of layer thickness.

**Figure 11 polymers-13-00556-f011:**
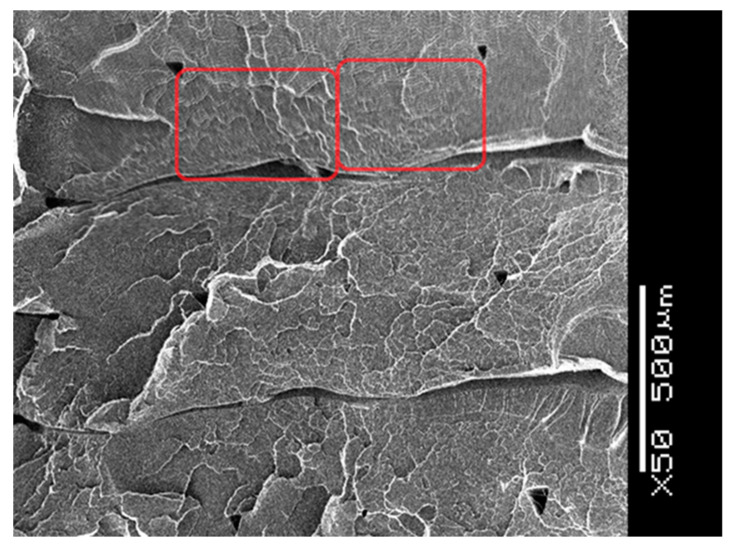
SEM picture of a specimen produced with 0.3 mm layer thickness.

**Figure 12 polymers-13-00556-f012:**
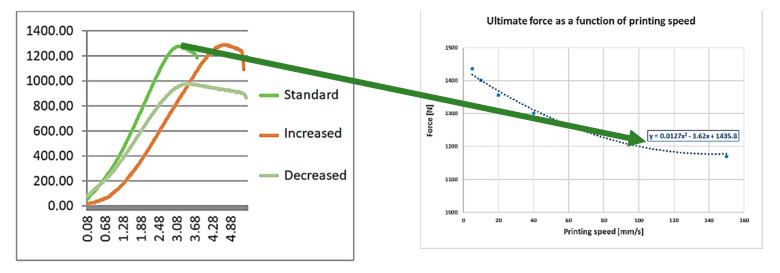
The tensile strength—printing speed function.

**Figure 13 polymers-13-00556-f013:**
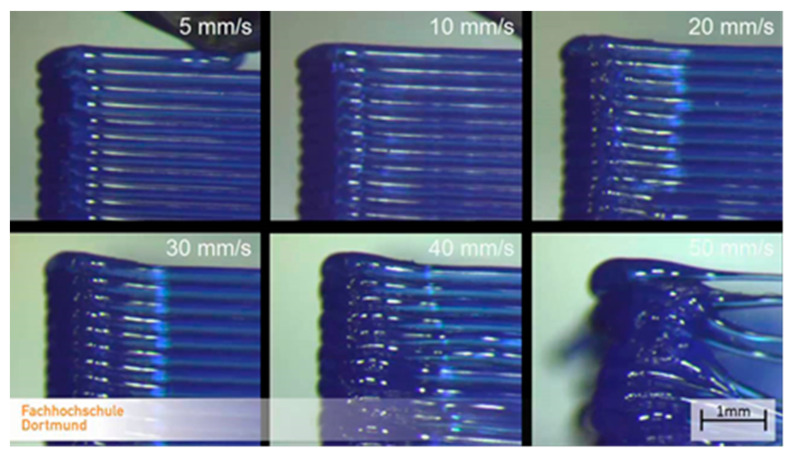
The speed effect on the model [43].

**Figure 14 polymers-13-00556-f014:**
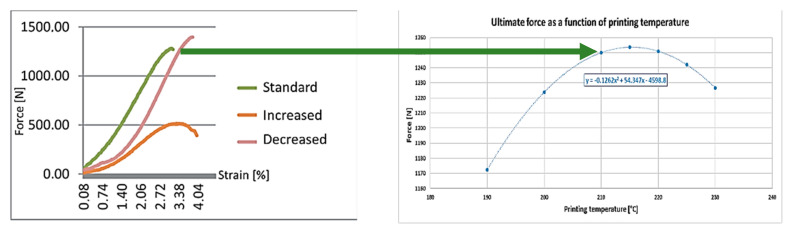
The tensile strength dependence on the printing temperature.

**Figure 15 polymers-13-00556-f015:**
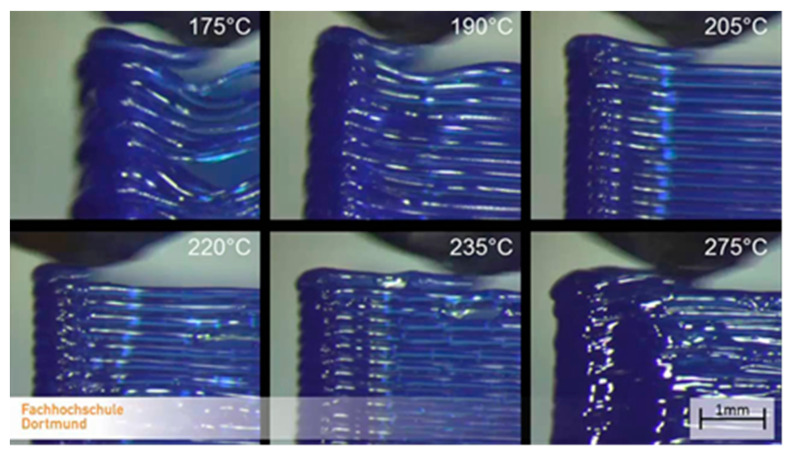
The temperature effect on the model [43].

**Table 1 polymers-13-00556-t001:** Production parameters of specimens.

No.	Temperature [°C]	Layer Thickness [mm]	Printing Speed [mm/s]
1.	200	0.1	40
2.	210	0.1	40
3.	215	0.1	40
4.	220	0.1	40
5.	225	0.1	40
6.	220	0.1	5
7.	220	0.1	10
8.	220	0.1	20
9.	220	0.1	60
10.	220	0.05	40
11.	220	0.2	40
12.	220	0.3	40
13.	220	0.4	40

## Data Availability

Data available on request due to restrictions eg privacy or ethical.

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
