# Peer review of "The Investigation of Interlaminar Failures Caused by Production Parameters in Case of Additive Manufactured Polymers"

_polymers, 2021, doi:10.3390/polym13040556_

Round 1

Reviewer 1 Report

This work concerns the investigation of interlaminar failures caused during 3D printing. Although there are enough experiments, there are some concerns concerning the presentation and their discussion. The title of the work constituted interlaminar failure but the fractorgraphs of the tensile samples were never presented nor discussed in the article.

Firstly, the abstract needs revision. There is no need to mention the sections within the abstract.

The references are not in order. Reference 1 is followed by reference 23. [line 25]
I am assuming that the author's reference manager had issues and this can be easily sorted out. Furthermore, the references are mixed with the first and last names. Unification is necessary for easier cross-reference and reading.

Line 27-28, authors mentioned: "for correct calculation". For calculation what exactly? It is not clear.

Line 41: fiber? or Filament? 

The introduction of the article does not provide enough information concerning the novelty of this work. The authors introduced polymer printing and fiber-reinforced polymer printing together, which is not mentioned again in the manuscript. 

In fact, a similar study already exists where a detailed investigation on the printing speed, nozzle diameter, and filament diameter on mechanical properties is presented. https://onlinelibrary.wiley.com/doi/abs/10.1002/app.49038

Line 65 [MPa] ->MPa

Authors point out to Figure 2 while referring to the improper bonding of layers, however, in Figure 2, no such phenomenon is observed.

Line 82-83: What does lfailure indicate?

Line 106, Acknowledgements are usually put at the end.

Figure 5: the legend is practically illegible. 

Line 111-113: What exactly are failures? Are porosities considered failures? Then the terminology is not correct. 

Figure 7: why is there only one equation? It does not represent all three cases. Further explanation is lacking.

Figure 10: Tensile testing results are presented but the testing conditions, sample dimensions, and the number of specimens are not mentioned in the manuscript.

Furthermore, what exactly does "Normal" "Increased" and "Decreased" mean?

If it is just the layer thickness, what was the total thickness of the sample?

Figure 11: Scale and magnification of SEM image are missing.

Figure 14: The curve is reaching about 1400N whereas the plot on the right has a maximum value of 1260N. Can authors provide more insights into this discrepancy?

Figure 10,12,14: the trend equations are missing and the explanations are not satisfactory. Authors should bring in more information to concretely support their statements.

Figures 13 and 15 are sourced from other references and are never mentioned in the manuscript. It is unclear how these images are linked to this current study.

In the end, authors never even suggested the optimum printing parameters and no suggestions/recommendations were made for future studies. 

Reviewer 2 Report

The manuscript can be accepted after minor revision.

Reviewer 3 Report

In this paper, the authors investigated the effect of three printing parameters on the structure of the printed element. However, certain issues need to be resolved and clarified.

  1. Introduction must be improved highlighting the aim of the paper, explaining the difference between this article and previous research.
  2. Figure 5 is too blurry and needs to be redrawn; Some figures are not clear, like Figure 2, Figure 8, Figure 9.
  3. Please explain the detailed reasons why three printing parameters affect the structure of the printing element.
  4. Figure 1, Figure 4, Figure 6, Figure 12, Figure 13, Figure 15 should appear in the corresponding part in manuscript and some figures have no explanation.

Round 2

Reviewer 1 Report

Firstly, the manuscript should again go through a grammar check. 

Furthermore, I still do not think "failure" is the right term to use. Porosity is a more appropriate term.

line 60: If the exact type of material is not important, will the results be the same in the case of any PLA? There are several PLA variants that exhibit different viscosities and different MFI. The grade of the PLA or at least, the supplier should be provided. What is Bq?

Line 63: Again, it is just one of the values for a PLA. cannot generalize the properties. try to mention the range

Line 73: optimum printing parameters should be identified.

Line 90: if--> in

Table 1: the sample numbering is not uniform

Figures 7,8,9,10,12,14 are difficult to read. The font size and line thickness should be increased.

Line 130: degressively increasing? degressive: tending to descend or decrease[Merriam-webster]
The wording should be changed. Same in line 185

Furthermore, the three lines are actually not necessary, are they? They are just the standard deviations for the mean value?

Can authors present a 3D surface plot that would facilitate an easier understanding of the parameters influencing the tensile strengths? 

Instead of using other author's images, why can't authors use their own images if they too observed similar defects during printing?

Furthermore, for figure 13, reference 44 is wrongly cited.

Equation 2 what is F and A?  Force at the break? and what are the values of A? was A constant for all the cases?

Line 132: Just one statement was provided. Why are the failure lengths decreasing and increasing? what are the factors influencing them? aren't figures 9 and 14 contradicting each other? Figure14 shows an increase in properties however increase in failure length?

also, there are only 3 curves on the left side and 5 representative points on the right-side figure. How is it possible? can authors present all the curves?

Figure 12: If standard, increased and decreased refer to layer thickness, why are they again presented here? Why are the curves same in the figure 10 and 12?

Line 165: two thickness values were mentioned and which is the actual thickness. And what is the part thickness? 

Comments on the replies:

  1. Even if the tests from the other paper are taken in this study, still the standard deviations and the mean values of those tests should be provided to get an idea.
  2. The experimental part should at least be explained in this paper. Furthermore, the authors stated that the validation should be carried out on standard specimens (Line 163). If the standard specimens are no used in this study, can the results be the same in every case?

Still, the conclusions are not clear and concrete.

Author Response

Reviewer1

Firstly, the manuscript should again go through a grammar check. 

Professional translator has checked again.

Furthermore, I still do not think "failure" is the right term to use. Porosity is a more appropriate term.

The text has been modified accordingly.

Due to the nature of this phenomenon the failures can be named as porosity, but, since this phenomenon should not appear in a perfect case it may better to name as failures.

line 60: If the exact type of material is not important, will the results be the same in the case of any PLA? There are several PLA variants that exhibit different viscosities and different MFI. The grade of the PLA or at least, the supplier should be provided. What is Bq?

The text has been modified accordingly.

To perform this study for matrix material, a commercial PLA (Polylactic acid) filament – produced by Bq, a Spanish filament supplier - has been used. For this investigation the exact type of material was not important, since in our article just the nature of this phenomenon has been investigated. Properties of different materials surely can influence of this phenomenon, but its nature will be the same.  PLA is a thermoplastic, holocrystal-line polymer produced by polycondensation or fermentation of hydroxy propionic acid. It behaves as a rigid material at room temperature, with the tensile strength of 50 MPa, and breaking strain of 3-5% - according to the manufacturer [34].

In this paper three printing parameters were investigated. Layer height and printing speed have a great affect on surface quality and costs. The smaller the layer height is, the longer the manufacturing is. Furthermore, they may have effect on mechanical properties. Temperature is a serious point during polymer manufacturing. Some important proper-ties of polymers like Melt Flow Index (MFI), decomposition depends on temperature, which means optimum must also be known for high quality 3D printing.

Line 63: Again, it is just one of the values for a PLA. cannot generalize the properties. try to mention the range

This is the official data of the manufacturer.

Line 73: optimum printing parameters should be identified.

It can be optimized for several purposes. We determined how changes in layer thickness, as well as print speed, affect stiffness. In the range of operation, there is not an exact optimum value. The increasing the layer thickness and printing speed increases the extent of the porosity.   In the case of the printing temperature we determined the optimal value.  

Line 90: if--> in

The text has been modified accordingly.

Table 1: the sample numbering is not uniform

The table has been modified accordingly.

Figures 7,8,9,10,12,14 are difficult to read. The font size and line thickness should be increased.

Figures are modified accordingly. The font size and line thickness significantly increased in case of Figures 7,8,9,10,12,14. With zoom the figures are sharp and clean to be read.

Line 130: degressively increasing? degressive: tending to descend or decrease[Merriam-webster]
The wording should be changed. Same in line 185

The text has been modified accordingly.

Furthermore, the three lines are actually not necessary, are they? They are just the standard deviations for the mean value?

The three lines are necessary, the minimum and the maximum values are describing the sample, and very rarely have connection with deviation. In statistics the deviation is a measure of average difference between the observed value and the aritmetic mean. Therefore, the deviation is much smaller then the difference between maximum and minimum.

Can authors present a 3D surface plot that would facilitate an easier understanding of the parameters influencing the tensile strengths?

Very good idea, authors also thought about it. However, authors would like to kindly remind the reviewers about the models of which have more then 3 dimensions are very hard to visualise. In our case 3 input and 1 output parameters were definied, therefore our 4 dimensional model could be presented in 3 pieces of 3D diagrams (3D surfaces) or 6 pieces of 2D graphs. For better understanding authors have chosen to make the 2D graphs. Authors have only showed the connection between input parameters and output parameter, so only 3 pieces of 2D graphs were incorported:

• tensile strength - layer thickness

• tensile strength - printing speed

• tensile strength - printing temperature

An additional difficulty is that the scaling of different physical quantities on different axes greatly influences the nature of the surface.

As a conclusion authors kindly as the reviewers to accept the 2D graphs.

Instead of using other author's images, why can't authors use their own images if they too observed similar defects during printing?

Where we had our own figure we used it. All other source were indicated.

Furthermore, for figure 13, reference 44 is wrongly cited.

The text has been modified accordingly.

Equation 2 what is F and A?  Force at the break? and what are the values of A? was A constant for all the cases?

F is Force at the break (calculated). A the real cross section (Theoretical cross section minus the air gaps (porosity) in the investigated cross section). These values ​​vary depending on the investigated printing parameters.

Line 132: Just one statement was provided. Why are the failure lengths decreasing and increasing? what are the factors influencing them? aren't figures 9 and 14 contradicting each other? Figure14 shows an increase in properties however increase in failure length?

There is no contradiction between figures 9 and 14. In both figures, it can be seen the same optimal value of the printing temperature.

Figure 14 shows an increasing properies up to the optimum point. Figure 9 shows a decreasing failure length up to the optimal point. 

also, there are only 3 curves on the left side and 5 representative points on the right-side figure. How is it possible? can authors present all the curves?

The detailed explanation can be found in the text just below Figure 10. 

Figure 12: If standard, increased and decreased refer to layer thickness, why are they again presented here? Why are the curves same in the figure 10 and 12?

Figure 12 has been corrected. Unfortunately in the editing process we copied the wrong diagram on the left side of the figure.

Line 165: two thickness values were mentioned and which is the actual thickness. And what is the part thickness? 

The text has been modified accordingly.

Comments on the replies:

  1. Even if the tests from the other paper are taken in this study, still the standard deviations and the mean values of those tests should be provided to get an idea.
  2. The experimental part should at least be explained in this paper. Furthermore, the authors stated that the validation should be carried out on standard specimens (Line 163). If the standard specimens are no used in this study, can the results be the same in every case?

1. In the cited literature, unfortunately, only the average values appear with different printing parameters. The deviations or other statistical parameters were not available.

2. The text has been modified accordingly.

Still, the conclusions are not clear and concrete.

The text has been modified accordingly.

Reviewer 3 Report

Can be accepted.

Author Response

Many thanks for your kind reply accepting our paper.

Please find the upgraded version.

Round 3

Reviewer 1 Report

Thank you for meticulously answering my review.

Author Response

Dear Reviewer,

Thank you for your comments.

The manuscript has been corrected according to your instruction.
